# Antifungal Activity of Industrial *Bacillus* Strains against *Mycogone perniciosa*, the Causative Agent of Wet Bubble Disease in White Button Mushrooms

**DOI:** 10.3390/microorganisms11082056

**Published:** 2023-08-10

**Authors:** Irina Novikova, Julia Titova

**Affiliations:** Federal State Budget Scientific Institution “All-Russian Research Institute of Plant Protection” (FSBSI VIZR), 3, Podbelskogo shosse, St. Petersburg 196608, Russia; irina_novikova@inbox.ru

**Keywords:** *Mycogone perniciosa*, industrial *Bacillus subtilis* strains, microbiocontrol, biological efficacy, multirecycled SMS aqueous extracts, submerged multibiorecycled biologics

## Abstract

White button mushrooms yield instability in artificial cultivation, often caused by crop diseases. The main disease is wet bubble disease. The appearance of its causative agent, *M. perniciosa*, may lead to total yield loss. The microbiocontrol of *M. perniciosa* is focused on casing soil antagonist use. Since no industrial producer strains of polyfunctional biologics have been used in previous studies, our research goal was to characterize the effect of *B. subtilis* B-10 and M-22 on a mycopathogen and reveal its control possibilities. The reason for *B. subtilis* B-10 and M-22 use in mycopathogen control has been revealed by interactions between producer strains and the studying of *M. perniciosa*. The suppression of *M. perniciosa* development by producer strains was established, indicating a prolonged B-10 and M-22 effect on the mycopathogen. High biological efficacy in both strains at the early stages of mycopathogen development upon introducing them into the wells and spraying was shown: B-10—50.9–99.6% and M-22—57.5–99.2%, respectively (*p* ≤ 0.05). Significant differences between producer strains were not revealed, although during the first day of exposure to developed *M. perniciosa* colonies, M-22 showed greater activity. The high efficiency of preventive treatment when producer strains completely suppressed mycoparasite development permits us to recommend them both for introducing when preparing casing for *M. perniciosa* control.

## 1. Introduction

White button mushroom *Agaricus bisporus* (J.E. Lange) Imbach has been artificially cultivated for over 450 years and has become one of the major crops [1,2,3]. The main problem in its farming is its yield instability. Even when adhering to every approved technological rule, the grower often cannot predict with complete certainty the optimal yield. One reason for this phenomenon is crop diseases, the harmfulness of which is still high in intensive cultivating technologies [4,5]. In this regard, the studying of the organisms responsible for white button mushroom yield losses is not losing relevance in terms of the scientifically motivated development of measures for their control [6]. The science and practice of mushroom farming has developed certain preventive measures to reduce the harm caused to mushroom cultures. The main measures, used both in our country and abroad, are compliance with compost and casing soil preparative technologies; adherence to hygiene requirements when performing work in mushroom houses; use of disease-resistant *A. bisporus* strains; use of the preventive measures of regular pre-sawing and inter-cycling in cultivation facilities; nutrient and hormone introduction; compliance with microclimate technological parameters at all stages of culture development; and the use of a limited chemical set to protect mushrooms [7,8]. Nevertheless, the full implementation of these complex measures often does not lead to the desired results. In the mushroom production process, optimal and stable yields cannot be ensured precisely because of crop diseases, caused mainly by micromycetes. The main representative of the white button mushroom mycopathogenic complex is the wet bubble causative agent *M. perniciosa* Magn. [9,10]. The disease manifests as basidiomata deformation with subsequent bacteria development as a secondary infection, causing intense fruit body rotting [4,9,10]. The cap initially shows a brown exudative blotch; later, the basidioma surface is covered with velvety white mycelium, and the fruit body decays. With slight lesions, an irregularly shaped cap with a swollen agaricaceous hymenophore is formed on the shortened stipe [4,10]. This disease occurs in industrial mushroom-growing farms almost everywhere when conditions are favorable for its emergence and development: temperatures of 16–20 °C, and humidity over 90% [5,7].

*M. perniciosa* has high viability. In culture, the micromycete retains its vital functions for 18 months; in nature, its spores retain viability for up to 3 years; when thermally treated, they withstand a temperature of 60 °C for more than 2 h; the mycelium dies within 30 min at this temperature. *M. perniciosa* and *A. bisporus* develop at almost the same temperature and pH ranges, but these ranges are somewhat wider for the mycoparasite (20–28 °C, pH 5–7) than for the host (22–24 °C, pH 5.5–5.6) [11].

Most often, the disease occurs at the end of the growing cycle, when the main crop is weakened and depleted, when mushroom beds are no longer adequately maintained, ventilation deteriorates, and the number of insect pathogen vectors increases [4]. Wet bubble disease is the most harmful disease of cultivated mushrooms. Its appearance often leads to entire crop loss in the cultivation chamber [4]. In its metabolism, *M. perniciosa* acidifies the habitat, which helps it develop resistance to chemicals, particularly benomyl [12,13].

*M. perniciosa* is an optional biotroph [14]. Its spores germinate on the *A. bisporus* mycelium in the casing, and from there, the mycopathogen’s hyphae penetrate into the fruit body anlage, and thus into the basidioma base. Mycopathogen spores can also germinate on already formed fruit bodies, but in this case, they do not cause serious damage to the host cells. On dead *A. bisporus* mycelium, *M. perniciosa* spores germinate well and the micromycete actively develops; thus, the primary infection occurs through mycelial strands, especially through dead ones left after basidiomata harvesting. From there, the parasite mycelium passes to the living mycelial pulls, and then to the generative formations—that is, the sclerodermoid infestation form. The common infestation form (the infestation of basidiomata) does not cause fruit body deformation; they stay covered with white *M. perniciosa* mycelium. Such an infestation form is very rare. It is important to note the lack of correlation between basidiomata resistance and the antagonistic activity of their mycelium against *M. perniciosa*. The infection persists in the casing in the form of chlamydospores [15,16]. The mycopathogen conidia and chlamydospores usually do not persist in compost because temperatures reach 70–80 °C during the compost processing, but the source of infection can also be compost if phytosanitary rules are violated during its preparation and transportation [17].

Due to its morphology and ecological and physiological characteristics, *M. perniciosa* spreads perfectly from the source of infection via air currents, water splashes, and sciarid flies, which carry off both light, small verticilliform conidia and strongly ornamented chlamydospores. Possessing mycophilic properties, the causative agent of wet bubble disease in white button mushroom is stimulated to develop by metabolites of the growing *A. bisporus* mushroom mycelium. The time between infestation and symptom expression usually varies from 2–3 to 12 days [18,19].

The microbiocontrol of the white button mushroom wet bubble disease causative agent has not yet competed with chemicals authorized for use in champignon farming [12]. The most actively investigated topics are the ability of biologics to influence the microbiota that determine the deterioration of the basidioma habitus, as well as the mushrooms’ competitive biota, mainly from micromycetes with high toxin-forming activity [13,20]. So far, only collected *Bacillus* strains isolated from casing soil and possessing high antagonistic activity have been estimated to control white button mushroom diseases [20,21]. Thus, the microbiological control of white button mushroom diseases is not popular in its industrial cultivation, despite the fact that the use of natural protective mechanisms, manifesting in regulating pest population numbers via their antagonists, has recently become a priority in agricultural crop protection. Agroecosystems phytosanitary optimizing involves the use of microorganisms with high ecological plasticity and various BAS complexes as polyfunctional biologics producer strains with complex biological activity [22]. For the development of polyfunctional biologics, the most interesting microbes are those that are non-pathogenic for humans and beneficial organisms, capable of producing a wide range of secondary metabolites, with low demand for their cultivation conditions, their high producibility, and their great ecological adaptability. The development of such technology comprises a number of stages, the foremost of which is the selection and optimization of nutrient substrates. Further, it is necessary to optimize the physicochemical conditions for cultivation and to develop methods to obtain the finished product, the quality of which significantly determines the duration of storage and the end-use biological activity of such biologics [22]. *Bacillus* bacteria, unpretentious with respect to growth conditions, their high-tech suitability, their ability to produce many BASs, and their ability to regulate the pathogen population density when introduced into the agrobiocenosis for a long time, meet such requirements [22].

*B. subtilis* strains B-10 and M-22 are highly effective against a wide range of various cultivated plant phytopathogens and have been deposited and passported at the Federal State Budget Scientific Institution “All-Russian Research Institute of Plant Protection” (FSBSI VIZR) Microbiocontrol laboratory [22]. The research–industrial technologies for polyfunctional biologics preparing on their basis were developed, and currently, four biologics included in the State catalog of pesticides and agrochemicals are permitted for use in the Russian Federation territory (2023) [23]. Based on *B. subtilis* B-10 b—Alirin-B, liquid (titer not less than 10^9^ CFU/mL); Alirin-B, tablet (titer not less than 10^9^ CFU/g); Alirin-B, wettable powder (titer not less than 10^11^ CFU/g); on *B. subtilis* M-22—Gamair, suspension concentrate (titer 10^10^ CFU/mL). Biologic application permitting is due to its inclusion into the State Catalog; after comprehensive series toxicological examining, both the producer strain and its biologic formulation are assigned to one of four danger classes for humans and warm-blooded animals. In addition, a toxicological examination is performed to establish the hazard to beneficial insects (bees) and fish [23]. The studied industrial strains *B. subtilis* B-10 and M-22 as well as their biologics formulations are safe for all objects included in the toxicological expertise [23]. These biologics have high biological efficacy (60–90%) against pathogens of the most harmful diseases, and their formulations include live cells and metabolome that determine the antagonistic activity as well as polypeptide antibiotics of original structures [22].

From the above, the research goal was to characterize the effect of *B. subtilis* B-10 and M-22 producer strains on *M. perniciosa* and evaluate the possibility of wet bubble control using antagonist microbes, the basis of polyfunctional biologics. To achieve this goal, the following tasks were solved: to compare the effect of *B. subtilis* B-10 and M-22 producer strains on 1-day- and 7-day-old *M. perniciosa* colonies when applied via spraying into the agar wells (methods simulating basidioma and casing soil treatments); to identify the most active antagonist to *M. perniciosa*; and to evaluate the ability of the studied producer strains to suppress the wet bubble disease development.

## 2. Materials and Methods

### 2.1. Materials Used in the Research

The research was carried out at the FSBSI VIZR Microbiocontrol laboratory and at its edible mushrooms’ pilot farm. Materials used for research were the spent mushroom substrates (SMSs) after industrial cultivating of *Lentinula edodes* (Berk.) Pegler (SMSLe)—shiitake mushrooms; and *Pleurotus ostreatus* (Jacq.) P. Kumm. (SMSPo)—oyster mushrooms, which were kindly provided by the edible mushroom farms “Prinevskoe” Ltd. and “Digris” Ltd. (Russia). There were used, multirecycled SMSs left after two edible mushroom species cultivating on the same substrate (SMSLePo). Substrates containing *L. edodes* 4080 and *P. ostreatus* NK-35 mycelium derivates were used as aqueous extracts for the obtaining of biologics producer strains from liquid culture [24]. The SMS was preliminarily disintegrated and boiled for 1 h in a 200 g/L volume, filtered, then restored to its original volume. Sterilization mode was 30 min at 81.1 kPa in autoclave 5075ELVPV D (Tuttnauer Europe BV, The Netherlands) [25]. Inoculation was carried out with submerged strains’ cultures in sterility conditions. Incubation was carried out at 26–28 °C. The trials were performed in vitro using Czapek-Dox agar (HiMedia Laboratories, India) [24,25].

### 2.2. Research Objects

The research objects were strains *B. subtilis* (Ehren.) Cohn B-10 and M-22—industrial polyfunctional biopesticide producers having antifungal activity, deposited and maintained in the State Collection of Microorganisms Pathogenic for Plants and Their Pests FSBSI VIZR. The Collection was registered on 28.01.1998 as No. 760 in the World Federation for Culture Collections, World Data Center for Microorganisms (WFCC WDCM, Japan). The research objects were the *B. subtilis* B-10 and M-22 liquid cultures obtained via the submerged fermenting in SMSLePo aqueous extracts. The production titers liquid cultures were *B. subtilis* B-10—3.2 × 10^12^ colony-forming units per 1 mL (CFU/mL) and B. subtilis M-22—1.2 × 10^12^ CFU/mL.

As a research object *M. perniciosa* was isolated from the casing soil at the 2^d^ fructification wave of *A. bisporus* X-22 in the “Prinevskoe” Ltd. farm (Moscow, Russia) (Figure 1).

### 2.3. Research Methods

The following standardized [26] as well as author-modified research microbiological methods [25] were used. The following spore productivity calculated parameter is presented: titer as viable CFU/mL [25,26]. The liquid-phase inoculum of the producer strains was grown at 27–28 °C in 3 days with aeration (150 rpm, New Brunswick™ Innova^®^ 44 incubator shaker, Eppendorf, Germany). The quantity and quality of the producer strains’ viable CFU/mL in the produced inoculums (titers) and the producer strains’ spore productivity was determined via a serial dilutions method on agar media via inoculating and spawning on Czapek-Dox agar (HiMedia Laboratories, India) and peptonic Czapek’s and Dried nutrient agar (Microgen Co., Ltd., Russia) [26].

*M. perniciosa* 1-day- and 7-days-old lawns were treated using submerged cultures of strains *B. subtilis* B-10 and M-22 using the well method and spraying with a laboratory sprayer. The *M. perniciosa* colony area as well as the micromycete’s lysis zone under producer strains treatment were determined using the circle area formula, where the radius is the colony or lysis zone radius. The antagonist microbes’ biological effectiveness (%) was determined using the mycelial lysis zone proportion in the total area occupied by the micromycete mycelium.

### 2.4. Statistical Processing Methods

Statistical obtained results processing using the Microsoft Excel 2010 and Statistica 10.0 software packages (StatSoft, Inc., Tulsa, OK, USA) included checking the normal distribution of the analyzed data with the help of the Shapiro–Wilk’s *W*-test, analysis of variance (ANOVA), and calculating the means (M) as well as using a standard errors (±SEM) calculation. The significance of the statistical differences in the options-pairwise comparison was assessed using Student’s *t*-test [27,28].

## 3. Results

### 3.1. Antifungal Activity of B. subtilis Strains against 1-Day-Old M. perniciosa Mycelium

After spraying a 1-day-old *M. perniciosa* lawn with *B. subtilis* B-10 and M-22 liquid cultures, no mycopathogen development was observed; on the 4th day intensive antagonist microbe development forming the basis of the biologics was detected. By the 8th cultivation day, the strain-producer lawn and complete *M. perniciosa* growth absence were observed (Figure 2).

### 3.2. Antifungal Activity of B. subtilis Strains against 7-Day-Old M. perniciosa Mycelium

When spraying 7-day-old *M. perniciosa* colonies with *B. subtilis* B-10 and M-22 liquid cultures, weak micromycete development was observed only during 3 days after treatment, and *M. perniciosa* colony lysis manifestation was registered on the 4th day. By the 8th cultivation day, the producer strains developing in the free part of the nutrient medium and an *M. perniciosa* lysis zone increase were observed (Figure 3).

The average rate of micromycete colony area increase in the control without treatment with antagonists was almost 30-fold higher than that after spraying 7-day-old colonies with *B. subtilis* B-10 liquid culture, and 16-fold higher after spraying with the *B. subtilis* M-22 strain. Moreover, treatment with the *B. subtilis* B-10 strain produced an almost two-fold decrease in the *M. perniciosa* growth rate compared to the decrease caused by *B. subtilis* M-22 spraying (Figure 4a). The average rate of *M. perniciosa* mycelial lysis increase under the *B. subtilis* B-10 effect was reliably (*p* ≤ 0.1) 1.2 times higher than that caused by the *B. subtilis* M-22 (Figure 4c). These dynamics of the indices during 15 days of mycopathogen cultivating after the spraying of the 7-day-old colonies with *B. subtilis* B-10 and M-22 liquid cultures is presented in Figure 4b,d, respectively. The introduction of antagonist microbes already after one day leads to *M. perniciosa* growth retardation, abundant sporulating in the colony marginal zone, and later to complete growth cessation, with sterile mycelial roll-forming as well as a barrage zone. Subsequently, these processes lead to extensive *M. perniciosa* mycelial lysis zone forming and the latter increasing during cultivation. The higher integral values of the lysis area increase rate function when exposed to *B. subtilis* B-10 are noteworthy, although the rate of *M. perniciosa* lysis area increase when exposed to *B. subtilis* M-22 has a peak exceeding by 1.7 times the same function value for *B. subtilis* B-10 (Figure 4d). The biological value of this phenomenon is that *B. subtilis* M-22 application causes a rapid and vast degradation of the *M. perniciosa* response, expressed in the rapid micromycete mycelium death and the impossibility of its development in the antagonist space. Exposure to *B. subtilis* B-10 leads to permanent growth retardation and *M. perniciosa* mycelium lysis, providing greater efficiency in suppressing the micromycete over time (Figure 4d).

### 3.3. Antifungal Activity of B. subtilis Strains Secondary Metabolites against M. perniciosa

The producer strains introduced into the wells on the 1-day-old *M. perniciosa* lawn resulted in weak micromycete development, leading to thin web-like mycelium with very few chlamydospores. By the 8th day the maximum zone of *M. perniciosa* lysis was formed around the wells on the poorly developed lawn (Figure 5a,d). The lysis proportion in the total area occupied by the mycelium was reliably (*p* ≤ 0.01) 2.3 times higher than that when the antagonists’ secondary metabolites were exposed to the formed *M. perniciosa* colonies. The differences between the *B. subtilis* B-10 or M-22 secondary metabolites effects were within the measurements error, although M-22 was more active as the zone of the *M. perniciosa* mycelium lysis was up to 4 mm larger than that caused by the B-10 secondary metabolites (Figure 5a,d). When producer strains were introduced into the wells and onto the developed *M. perniciosa* lawn, their growth was observed directly in the wells, without coming onto the developed *M. perniciosa* lawn. No lysis zones were observed, but the producer strains continued to develop in the wells of the *M. perniciosa* lawn (Figure 5b,c,e,f).

Liquid form spraying of any from studied producer strains on *M. perniciosa* colonies at very early stages of mycopathogen development led to its development complete slowing and death. The exometabolome mycocidal activity against the pathogen was 61.7% for *B. subtilis* B-10 and 50.1% for *B. subtilis* M-22, respectively. A one-week delay in treatment via spraying reduced the mycocidal activity of both producer strains by more than 3 times. Local application of the antagonists by introducing them into wells at early stages of *M. perniciosa* development revealed the producer strain secondary metabolites mycocidal biological activity level for *B. subtilis* B-10—68.6% and *B. subtilis* M-22—59.0%, respectively. The differences in the antagonists’ activity were insignificant and reliable: *B. subtilis* M-22 more actively suppressed the mushroom wet bubble pathogen. *B. subtilis* B-10 and M-22 local application via introduction into wells on the sporulating developed *M. perniciosa* lawn was ineffective.

## 4. Discussion

Active *Bacillus* strains have long been widely used in microbiologicals having various action spectrums because of their polyfunctionality [21]. Bacilli synthesizes a wide range of antimicrobial compounds that belong to various chemical classes. In particular, *B. amyloliquefaciens* F9 (F9LE) has been found to produce three lipopeptides types, including homologs of surfactin, fengycin, and iturin, that inhibit the citrus cancer pathogen growth caused by *Xanthomonas citri* subsp. *citri* [29]. *B. amyloliquefaciens* VB7 has been shown to be a potential antagonist that can control a number of dangerous diseases (*Sclerotinia sclerotiorum* affecting cloves, tobacco stripe virus affecting cotton, and peanut bud necrosis affecting tomatoes) and synthesizes non-ribosomal antimicrobial peptides such as surfactin, butirosin A/butirosin B, fengycin, difficidin, bacillibactin, bacilysin, and mersacidin the Ripp lanthipeptide, as well as a number of other compounds as well [30].

Recently, various mechanisms of phytopathogen suppression by *Bacillus* strains have been actively studied [30,31,32,33]. In particular, chitinase and β-1,3-glucanase enzymes produced by *B. velezensis* CE 100 and the effectively lysing mycelium of *Macrophomina phaseolina* and *Fusarium oxysporum* f. sp. *Fragariae* have been found to cause charcoal rot and the wilting of strawberries [34]. Representatives of the genus *Bacillus* have not only a direct antagonistic effect on phytopathogenic species, but also stimulate mechanisms for increasing plant disease resistance. For example, *Bacillus velezensis* BY6 significantly increased the expression levels of defense genes related to salicylic and jasmonic acid signaling pathways, simultaneously activating induced systemic resistance and systemic acquired resistance in diseased poplar seedlings [35]. The evaluation of several strains of *B. subtilis*, *B. pumilus*, and *B. megaterium* in protecting celery against powdery mildew caused by *Erysiphe heraclei* DC showed high efficacy in the suppression of disease, with significant improvement in growth and yield characteristics. In addition, they caused an increase in leaf pigment concentration, activities of the defense enzymes peroxidase and polyphenol oxidase, and total phenolic content [36].

The biomethod for the control of cultivated white button mushroom diseases has not yet found wide application in industrial cultivating. Biologics based on *Bacillus* strains are mainly used in modern *A. bisporus* growing to control its competitive microbiota due to the microcidal and microstatic activities of its metabolome [37,38,39,40]. Only some approved chemicals are used to control mycoparasite *M. perniciosa* [12,41,42]. The search for microbiocontrol possibilities of wet bubble pathogen is focused on antagonists isolating and use from the casing soil to which *M. perniciosa* shows sensitivity. [20,21,22,23,24,25,37,38,39,40,41,42,43]. These studies do not use highly active industrial producer strains of polyfunctional biologics with fungicidal activity, which are *B. subtilis* B-10 and M-22 [21]. Active secondary metabolites produced by *B. subtilis* B-10 and M-22 were isolated and characterized using extraction and chromatographic and spectrometric methods (UV-, IR-, NMR (^1^H and ^13^C), mass spectrometry) [44]. The active complex of *B. subtilis* B-10 includes polypeptide and polyene antibiotics. The main antibiotic Alirin B_1_ was classified as bacteriocin. The antibiotic physicochemical and biological properties as well as the amino acid composition were characterized; significant differences between Alirin B_1_ and the polypeptide preparations described earlier in the literature were shown, and its structure originality was proven [45]. The broad spectrum of *B. subtilis* M-22 antagonistic activity is due to the active complex component composition analyzed via HPLC with UV and mass spectrometric detection. This complex includes various chemical compounds: gamair A is a polypeptide close to bacillin, gamair B is a hexaene antibiotic classified as subgroup 1A (mediocidin), and gamairs C and D are also hexaene antibiotics [46]. According to the data obtained, the chemical nature of other active components as carbohydrates high-molecular-weight was predicted with high probability [44,46].

The interaction between the above producer strains *B. subtilis* B-10 and M-22 and the well-developed *A. bisporus* mycelium in the transition stage (popcorn-like generative forms appearing) does not negatively affect morphogenesis [20,23]. Furthermore, the new approaches to the production of polyfunctional biologics via the multibiorecycling of agricultural and industrial wastes have been developed based on a cenotic conception of biorecycling as an extension of the food chain and an expansion of the nutrition supply network [24,25]. New growth substrates have been developed to facilitate the development of producer strains for biologics use on agro-industrial wastes. Such substrates can be repeatedly and effectively recycled, with edible macromycetes producing 60–70% of commercial fruit body yields [24]. New data have been obtained on the residual nutritional value of waste substrates remaining after commercial mushroom growing, resulting in a 1.4- and 2.2-fold decrease in the content of lignin and cellulose compared to their initial levels. In the process, hard-to-degrade polymers of the lignocellulose complex are converted into assimilable water-soluble forms as the result of successive enzymatic degradation. Enrichment of converted substrates led to 3.8 and 3.5 times more total nitrogen and crude protein, up to 3.5 times more vitamins, about 2 times more micronutrients, and a more-than-3-times-less C/N ratio [24]. The high efficiency of the multibiorecycling of waste substrates that remain after cultivation of edible macromycetes has been shown by *Bacillus* industrial strains [24,47]. The feasibility of obtaining environmentally friendly biologics using SMS extracts has been shown [25,47]. Storage conditions for experimental formulations have been optimized, ensuring preservation of viability and target activity of the producer strains. Such novel formulations can provide opportunities for a more complete and more efficient implementation of phytosanitary traits of biologics in vivo via optimizing the microbial complex in casing and *A. bisporus* hyphospheres [23,24,25,47].

## 5. Conclusions

The use of *B. subtilis* B-10 and M-22 producer strains to control *M. perniciosa* development has been revealed. Significant differences in the micromycete colonies’ growth rate suppression in terms of its mycelium lysis zone area were detected. All trials’ variants have shown a positive dynamic in the suppresion of *M. perniciosa* development, which indicates a prolonged *B. subtilis* B-10 and M-22 effect on the *A. bisporus* wet bubble disease causative agent. The high biological efficacy of both producer strains against *M. perniciosa* at the early stages of its development by introducing into the wells and spraying was shown: *B. subtilis* B-10—50.9–99.6% and *B. subtilis* M-22—57.5–99.2%, respectively (*p* ≤ 0.05). Significant differences between the producer strains were not revealed, although during the first day of exposure to the developed *M. perniciosa* colonies *B. subtilis* M-22 showed greater activity: the zone of *M. perniciosa* mycelium lysis was up to 4 mm larger than that caused by B-10 activity. The discovered high efficiency of the preventive treatment when the producer strains completely suppressed the *M. perniciosa* development by their secondary metabolites made it possible to recommend *B. subtilis* B-10 and M-22 introducing when preparing casing to control mycopathogen development.

## Figures and Tables

**Figure 1 microorganisms-11-02056-f001:**
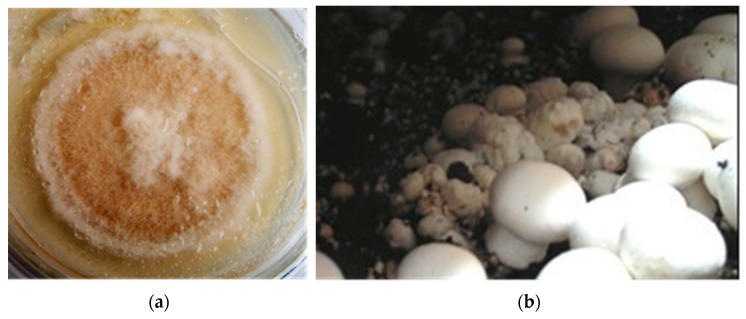
*M. perniciosa* pure culture (**a**) and *A. bisporus* fruit bodies infection (**b**).

**Figure 2 microorganisms-11-02056-f002:**
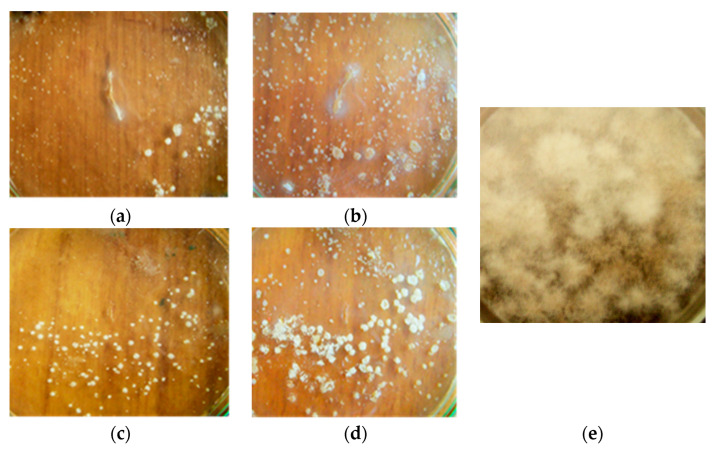
*M. perniciosa* development after inoculating and simultaneous spraying with *B. subtilis* B-10 (**a**,**b**) and M-22 (**c**,**d**) producer strain liquid cultures: (**a**,**c**)—4th cultivation day; (**b**,**d**)—8th cultivation day; (**e**)—control *M. perniciosa* without treatment, 8th cultivation day.

**Figure 3 microorganisms-11-02056-f003:**
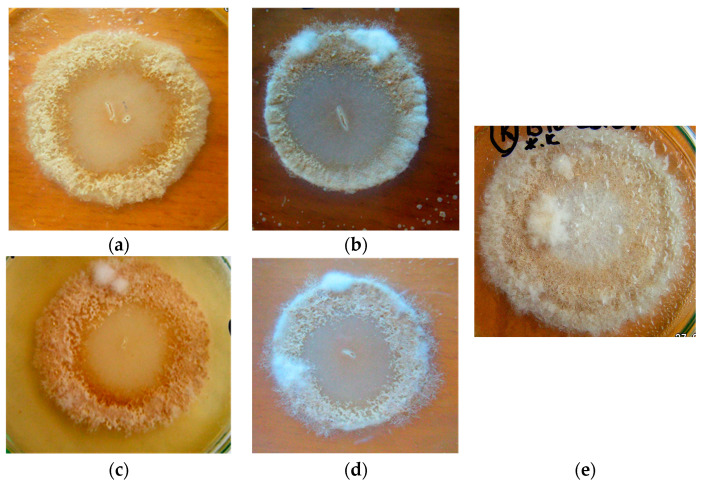
*M. perniciosa* development after spraying its 7-day-old colonies with *B. subtilis* B-10 (**a**,**b**) and M-22 (**c**,**d**) producer strains liquid cultures and (**e**) sterile water (control): (**a**,**c**)—4 days after treatment; (**b**,**d**,**e**)—8 days after treatment.

**Figure 4 microorganisms-11-02056-f004:**
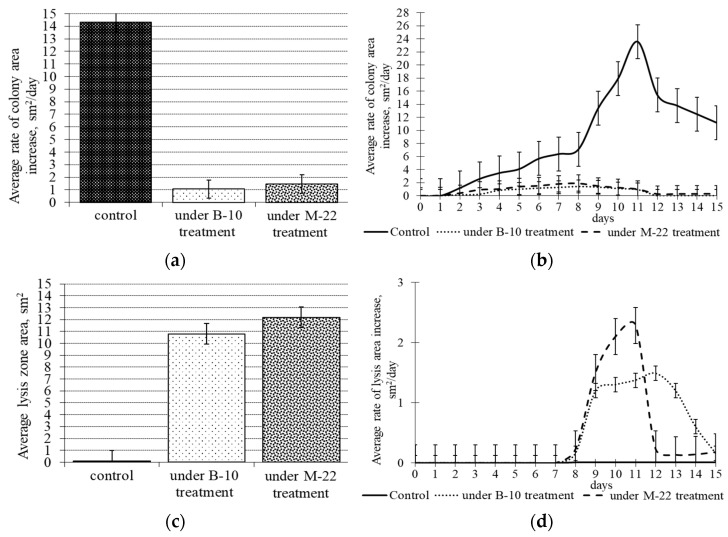
Parameters of development and mycelium lysis zone formation of *M. perniciosa* during 15 days of cultivating after 7-day-old colonies were sprayed with *B. subtilis* B-10 and M-22 producer strain liquid cultures: (**a**)—average increase rate of colony area; (**b**)—average colony area dynamics; (**c**)—mycelium lysis zone; (**d**)—mycelium lysis zone dynamics.

**Figure 5 microorganisms-11-02056-f005:**
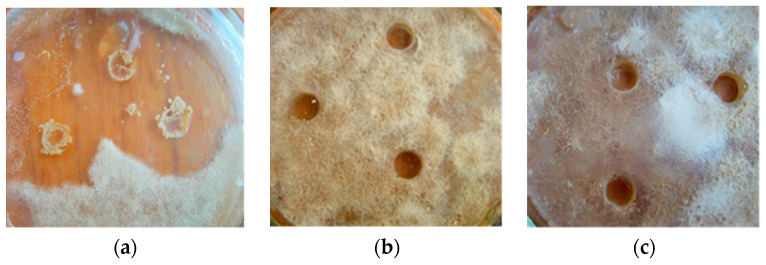
*M. perniciosa* mycelium lysis zone development after placing into agar wells of *B. subtilis* B-10 (**a**–**c**) and M-22 (**d**–**f**) producer strain liquid cultures: (**a**,**d**)—simultaneously with *M. perniciosa* inoculating; (**b**,**c**,**e**,**f**)—on 7-day-old *M. perniciosa* lawn; (**b**)—B-10; 4 days after putting into wells; (**a**,**c**)—B-10, 8 days after putting into wells; (**e**)—M-22, 4 days after putting into wells; (**d**,**f**)—M-22, 8 days after putting into wells.

## Data Availability

Not applicable.

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
