# Peer review of "Antifungal Activity of Industrial Bacillus Strains against Mycogone perniciosa, the Causative Agent of Wet Bubble Disease in White Button Mushrooms"

_microorganisms, 2023, doi:10.3390/microorganisms11082056_

Round 1

Reviewer 1 Report

Manuscript Title: Antifungal activity of industrial Bacillus strains against Mycogone perniciosa the causative agent of cultivated mushrooms wet bubble disease

Authors: Irina Igorevna Novikova, Julia Anatolievna Titova

 The work provides some interesting results concerning the Antifungal activity of industrial Bacillus strains against Mycogone perniciosa. Generally speaking, this paper was well organized and the data was properly testified. However, the following suggestions should be considered. 

 Major comments:

 There are many reports on the use of Bacillus strains for the biocontrol. It is suggested to add a comparison with the existing reported results of the biocontrol ability to manage bacterial and fungal pathogens using Bacillus strains at the end of the discussion to clarify the innovation of this study.

 Technical queries/ suggestions:

 1. Abstract should be written more precisely and explain novelty of this work. I suggest the abstract only writes the main results. Most of the methods should be deleted. Such as “Interactions between producer strains and M. perniciosa were evaluated using well method and spraying. Mycoparasite colony area and its lysis zone were determined. Biological efficacy was characterized by lysis zone proportion in total area occupied by the micromycete mycelium. Statistical processing included W-test, ANOVA, means (M) and standard errors (±SEM) calculation. The significance was assessed by t-test.” These sentences should be deleted.

2. The Introduction section should be improved. This section is too long at present. Introduction should briefly place the study in a broad context and highlight why it is important. It should define the purpose of the work and its significance. The current state of the research field should be reviewed, and key publications cited. Finally, briefly mention the main aim of the work and highlight the principal conclusions. However, the novelty and significance of the manuscript were not highlighted in the Introduction section, please modify the introduction more clearly.

3. In many places of work, there are no citations (e.g. lines 101-128, 132-138, and so on).

4. Line 287: 2.3. . Methods. This section should be written in subsection. Using Methods is too general. Which methods?

5. Statistical analysis is very important. I suggest the authors add a new section in the Materials and Methods to describe the details of the statistical analysis.

6. Line 276: Discussion. The poor discussion of the results. Author have showed the great amount of results that they have achieved, but they did not use them to develop an interesting discussion which could supplement to earlier studies on the biocontrol activity using Bacillus strains. Bacillus strains are known for their metabolic capability and environmental versatility as well as for their ability to manage bacterial and fungal pathogens. Authors should add more information into this section and cite the recent research into the field. However, the authors did not compare the biocontrol activity of the selected Bacillus strains with that of other Bacillus strains based on the literature such as Bacillus velezensis BY6 Promotes Growth of Poplar and Improves Resistance Contributing to the Biocontrol of Armillaria solidipes. Microorganisms. 2022; doi: Control of Fungal Diseases and Fruit Yield Improvement of Strawberry Using Bacillus velezensis CE 100. Microorganisms. 2022; Isolation and Screening of Antagonistic Endophytes against Phytophthora infestans and Preliminary Exploration on Anti-oomycete Mechanism of Bacillus velezensis 6-5. Plants (Basel). 2023; Isolation of the Novel Strain Bacillus amyloliquefaciens F9 and Identification of Lipopeptide Extract Components Responsible for Activity against Xanthomonas citri subsp. citri. Plants (Basel). 2022; Biological Control of Celery Powdery Mildew Disease Caused by Erysiphe heraclei DC In Vitro and In Vivo Conditions. Plants (Basel). 2021; doi: Control of postharvest green mold of citrus fruit by application of endophytic Paenibacillus polymyxa strain SG-6. Postharvest Biology and Technology; Mining the Genome of Bacillus velezensis VB7 (CP047587) for MAMP Genes and Non-Ribosomal Peptide Synthetase Gene Clusters Conferring Antiviral and Antifungal Activity. Microorganisms. 2021; Combination of hot water, Bacillus amyloliquefaciens HF-01 and sodium bicarbonate treatments to control postharvest decay of citrus fruit. Postharvest Biology and Technology.

7. Please check all the species names. Species names are typically given in full the first time they are used within the main text and then abbreviated throughout the remainder of the text.

8. Please check throughout the manuscript that abbreviations/acronyms are defined the first time they appear in each of three sections: the abstract; the main text; the first figure or table.

9. The Conclusion section can be shortened and written more precisely. This section should be revised for the better understanding of the topic and its future research.

English Language can be polished.

Author Response

Response to Reviewer 1 Comments

Point 1: There are many reports on the use of Bacillus strains for the biocontrol. It is suggested to add a comparison with the existing reported results of the biocontrol ability to manage bacterial and fungal pathogens using Bacillus strains at the end of the discussion to clarify the innovation of this study.

Response 1: the comparison with the existing reported results of the biocontrol ability to manage bacterial and fungal pathogens using Bacillus strains is added to the discussion.

Point 2: Abstract should be written more precisely and explain novelty of this work. I suggest the abstract only writes the main results. Most of the methods should be deleted. Such as “Interactions between producer strains and M. perniciosa were evaluated using well method and spraying. Mycoparasite colony area and its lysis zone were determined. Biological efficacy was characterized by lysis zone proportion in total area occupied by the micromycete mycelium. Statistical processing included W-test, ANOVA, means (M) and standard errors (±SEM) calculation. The significance was assessed by t-test.” These sentences should be deleted.

Response 2: Most of the methods are deleted from the abstract which contains now the novelty of our work and the main results.

Point 3: The Introduction section should be improved. This section is too long at present. Introduction should briefly place the study in a broad context and highlight why it is important. It should define the purpose of the work and its significance. The current state of the research field should be reviewed, and key publications cited. Finally, briefly mention the main aim of the work and highlight the principal conclusions. However, the novelty and significance of the manuscript were not highlighted in the Introduction section, please modify the introduction more clearly.

Response 3: the Introduction section is modified according to demands and wishes.

Point 4: In many places of work, there are no citations (e.g. lines 101-128, 132-138, and so on).

Response 4: the citations are included into the text.

Point 5: Line 287: 2.3. Methods. This section should be written in subsection. Using Methods is too general. Which methods?

Response 5: the title of 2.3 paragraph was corrected.

Point 6: Statistical analysis is very important. I suggest the authors add a new section in the Materials and Methods to describe the details of the statistical analysis.

Response 6: the new 2.4. section was added to describe the details of the statistical analysis.

Point 7: Line 276: Discussion. The poor discussion of the results. Author have showed the great amount of results that they have achieved, but they did not use them to develop an interesting discussion which could supplement to earlier studies on the biocontrol activity using Bacillus strains. Bacillus strains are known for their metabolic capability and environmental versatility as well as for their ability to manage bacterial and fungal pathogens. Authors should add more information into this section and cite the recent research into the field. However, the authors did not compare the biocontrol activity of the selected Bacillus strains with that of other Bacillus strains based on the literature such as Bacillus velezensis BY6 Promotes Growth of Poplar and Improves Resistance Contributing to the Biocontrol of Armillaria solidipes. Microorganisms. 2022; doi: Control of Fungal Diseases and Fruit Yield Improvement of Strawberry Using Bacillus velezensis CE 100. Microorganisms. 2022; Isolation and Screening of Antagonistic Endophytes against Phytophthora infestans and Preliminary Exploration on Anti-oomycete Mechanism of Bacillus velezensis 6-5. Plants (Basel). 2023; Isolation of the Novel Strain Bacillus amyloliquefaciens F9 and Identification of Lipopeptide Extract Components Responsible for Activity against Xanthomonas citri subsp. citri. Plants (Basel). 2022; Biological Control of Celery Powdery Mildew Disease Caused by Erysiphe heraclei DC In Vitro and In Vivo Conditions. Plants (Basel). 2021; doi: Control of postharvest green mold of citrus fruit by application of endophytic Paenibacillus polymyxa strain SG-6. Postharvest Biology and Technology; Mining the Genome of Bacillus velezensis VB7 (CP047587) for MAMP Genes and Non-Ribosomal Peptide Synthetase Gene Clusters Conferring Antiviral and Antifungal Activity. Microorganisms. 2021; Combination of hot water, Bacillus amyloliquefaciens HF-01 and sodium bicarbonate treatments to control postharvest decay of citrus fruit. Postharvest Biology and Technology.

Response 7: much more information is added to the Discussion section on the biocontrol activity using Bacillus strains.

Point 8: Please check all the species names. Species names are typically given in full the first time they are used within the main text and then abbreviated throughout the remainder of the text.

Response 8: all the species names and their abbreviations were checked.

Point 9: Please check throughout the manuscript that abbreviations/acronyms are defined the first time they appear in each of three sections: the abstract; the main text; the first figure or table.

Response 9: all the abbreviations/acronyms were defined the first time they appeared.

Point 10: The Conclusion section can be shortened and written more precisely. This section should be revised for the better understanding of the topic and its future research.

Response 10: the Conclusion section is shortened and revised.

Reviewer 2 Report

The manuscript, entitled “Antifungal activity of industrial Bacillus strains against Mycogone perniciosa the causative agent of cultivated mushrooms wet bubble disease” described that to characterize the effect of B. subtilis B-10 and M-22 producer strains on M. perniciosa and evaluate the possibility of wet bubble control using antagonist microbes, the authors evaluated the Interactions between producer strains and M. perniciosa. The results showed high biological efficacy of both producer strains against M. perniciosa at the early stages of its development. However, due to some drawbacks, my suggestion is major revision.

Comments:

1. Firstly, what do the authors think causes B-10 and M-22 to be able to inhibit the growth of M. perniciosa? Is it through competition between strains, or is it because some natural product synthesised by B. subtilis inhibits the growth of M. perniciosa? What are the specific reasons? The authors can first collect a culture of B. subtilis by incubating it and then spray the culture onto M. perniciosa to determine if it is the strain or the metabolites that are playing a role.

2. Are B. subtilis B-10 and M-22 safe for humans? Could residues of B. subtilis B-10 or M-22, which are sprayed on mushrooms to treat fungal diseases caused by Mycogone perniciosa, affect human health? If the metabolites of B. subtilis act as inhibitors of Mycogone perniciosa, the above question is resolved.

3.. From line 225 to 227, it is stated that “The average rate of micromycete colony area increase in the control without treatment with antagonists was almost 30-fold higher than that after spraying 7-d-old colonies with B. subtilis B-10 liquid culture and 16-fold higher after spraying with B. subtilis M-22 strain.”. How did the authors detect and caculate it.

4. The INTRODUCTION section is too redundant, and it is sufficient for the section to provide some basic background knowledge for the content of the study.

5. In the discussion section, the authors should try to analyze the data of the results and compare the data in other papers.

Author Response

Response to Reviewer 2 Comments

Point 1: Firstly, what do the authors think causes B-10 and M-22 to be able to inhibit the growth of M. perniciosa? Is it through competition between strains, or is it because some natural product synthesised by B. subtilis inhibits the growth of M. perniciosa? What are the specific reasons? The authors can first collect a culture of B. subtilis by incubating it and then spray the culture onto M. perniciosa to determine if it is the strain or the metabolites that are playing a role.

Response 1: all explanations now are given in the text.

Point 2: Are B. subtilis B-10 and M-22 safe for humans? Could residues of B. subtilis B-10 or M-22, which are sprayed on mushrooms to treat fungal diseases caused by Mycogone perniciosa, affect human health? If the metabolites of B. subtilis act as inhibitors of Mycogone perniciosa, the above question is resolved.

Response 2: all the explanations are given in 143–154 rows.

Point 3: From line 225 to 227, it is stated that “The average rate of micromycete colony area increase in the control without treatment with antagonists was almost 30-fold higher than that after spraying 7-d-old colonies with B. subtilis B-10 liquid culture and 16-fold higher after spraying with B. subtilis M-22 strain.”. How did the authors detect and caculate it.

Response 3 it was calculated from the ratio of the change rate in the micromycete colony area in the experiment and control.

Point 4: The INTRODUCTION section is too redundant, and it is sufficient for the section to provide some basic background knowledge for the content of the study.

Response 4: the Introduction section was shortened and revised.

Point 5: In the discussion section, the authors should try to analyze the data of the results and compare the data in other papers.

Response 5: the Discussion section was revised.

Reviewer 3 Report

Manuscript: „Antifungal activity of industrial Bacillus strains against Mycogone perniciosa the causative agent of cultivated mushrooms wet bubble disease“

Irina Novikova and Julia Titova 

The current research covers many topics associated with sustainable agriculture, phytopathology, and microbiological approaches. The main purpose of the study was to characterize Bacillus subtilis B-10 and M-22 the influence on the wet bubble causative agent Mycogone perniciosa. It is known, that Bacillus subtilis strains B-10, M-22 highly effective against a wide range of various cultivated plant phytopathogens. In the present research, the potential application of B. subtilis B-10 and M-22 producer strains to control the Mycogene perniciosa development has been analyzed.

Comments and suggestions to the authors:

1)      I think is too many keywords and they should not be similar to the title.

2)     It should be correct the writing of titles of 2.1; 2.2; 2.3 paragraphs.

3)    It should be correct the sentence: „The peculiarities of M. perniciosa morphology are following“ in the paragraph 85.

4)     It should be correct 16–20 °C; 60 °C; 20–28 °C; 22–24 °C, 70– 80 °C in the paragraphs 55; 58; 60; 61; 83.

5)     It should be correct pH 5–7; pH 5.5–5.6 in the paragraphs 60, 61.

6)      It should be correct „Here at“ in the paragraph 230;

7)     It should be correct Figure 4 b, d, in the paragraph 234. 

Author Response

Response to Reviewer 3 Comments

Point 1: I think is too many keywords and they should not be similar to the title.

Response 1: the number of keywords was reduced, the remaining ones are not similar to the article title.

Point 2: It should be correct the writing of titles of 2.1; 2.2; 2.3 paragraphs.

Response 2: the titles of 2.1; 2.2; 2.3 paragraphs were corrected.

Point 3: It should be correct the sentence: „The peculiarities of M. perniciosa morphology are following“ in the paragraph 85.

Response 3: the sentence was corrected.

Point 4:  It should be correct 16–20 °C; 60 °C; 20–28 °C; 22–24 °C, 70– 80 °C in the paragraphs 55; 58; 60; 61; 83.

Response 4: all the corrections were made.

Point 5: It should be correct pH 5–7; pH 5.5–5.6 in the paragraphs 60, 61.

Response 5: all the corrections were made.

Point 6:  It should be correct „Here at“ in the paragraph 230.

Response 6: the correction was made.

Point 7: It should be correct Figure 4 b, d, in the paragraph 234.

Response 7: the correction was made.

Round 2

Reviewer 1 Report

The manuscript has been improved and can be accepted.

Author Response

The reviewer 1 has no questions, comments or suggestions. 

Reviewer 2 Report

The authors of the manuscript named “Antifungal activity of industrial Bacillus strains against Mycogone perniciosa the causative agent of cultivated mushrooms wet bubble disease” have revised the text according to the comments. However, due to some drawbacks, my suggestion is major revision.

Comments:

1. Firstly, I think the study is meaningful for mushroom cultivation, but there is less research in this study and little explanation of the mechanism. In the last round of reviews, I asked why Bacillus subtilis was able to inhibit the growth of M. perniciosa? But the authors did not answer the question. I suggested that the authors should use techniques such as LC-MS to isolate and identify the main components in Bacillus subtilis and verify which component played a role, so as to deepen the depth of the article's research.

2. In addition, the authors were able to study which metabolic pathways changed in M. perniciosa before and after s spraying by means of histological tools such as RNA-seq. It is recommended that the authors add relevant experiments carefully, otherwise the manuscript cannot be published in a journal with the current research content.

3. The INTRODUCTION section is still too redundant, and it is sufficient for the section to provide some basic background knowledge for the content of the study.

4. The names of genus and species should be in italics, such as M. perniciosa and B. subtilis in abstract section.

The names of genus and species should be in italics, such as M. perniciosa and B. subtilis in abstract section.

Author Response

Response to Reviewer 2 Comments

Point 1: Firstly, I think the study is meaningful for mushroom cultivation, but there is less research in this study and little explanation of the mechanism. In the last round of reviews, I asked why Bacillus subtilis was able to inhibit the growth of M. perniciosa? But the authors did not answer the question. I suggested that the authors should use techniques such as LC-MS to isolate and identify the main components in Bacillus subtilis and verify which component played a role, so as to deepen the depth of the article's research.

Response 1: the explanations to why Bacillus subtilis was able to inhibit the growth of M. perniciosa are given in the 260, 273–297, 300–301, 305–306, 306–319, 358 rows.

Point 2: In addition, the authors were able to study which metabolic pathways changed in M. perniciosa before and after s spraying by means of histological tools such as RNA-seq. It is recommended that the authors add relevant experiments carefully, otherwise the manuscript cannot be published in a journal with the current research content.

Response 2: We dare remind the sincerely respected reviewer the goal and tasks of this study: “the research goal was to characterize the effect of B. subtilis B-10 and M-22 producer strains on M. perniciosa and evaluate the possibility of wet bubble control using antagonist microbes, the basis of polyfunctional biologics. To achieve this goal, the following tasks were solved: to compare the effect of B. subtilis B-10 and M-22 producer strains on 1-day- and 7-day-old M. perniciosa colonies when applied by spraying and into the agar wells (methods simulating basidioma and casing soil treatments); to identify the most active antagonist to M. perniciosa; to evaluate the ability of the studied producer strains to suppress the wet bubble disease development”. “To study which metabolic pathways changed in M. perniciosa before and after spraying by means of histological tools such as RNA-seq” hadn’t been the task of the present research.

Point 3: The INTRODUCTION section is still too redundant, and it is sufficient for the section to provide some basic background knowledge for the content of the study.

Response 3: The INTRODUCTION section is shortened deleting 84–91 rows (marked in green color).

Point 4: The names of genus and species should be in italics, such as M. perniciosa and B. subtilis in abstract section.

Response 4: all genus and species names in abstract section are written in italics now (marked in green color).

Reviewer 3 Report

Thanks authors  for the made corrections in  the revised manuscript. I don't have any comments or suggestions. 

Author Response

The reviewer 3 has no questions, comments or suggestions.